# Association between Quality of Life, Confinement, and Sex in Adults: A Multigroup Structural Equation Analysis

**DOI:** 10.3390/healthcare12070774

**Published:** 2024-04-03

**Authors:** Félix Zurita-Ortega, Dilan Galeano-Rojas, Dennys Tenelanda-López, Mauricio Cresp-Barria, Claudio Farias-Valenzuela, Pedro Valdivia-Moral

**Affiliations:** 1Department of Didactics Musical, Plastic and Corporal Expression, Faculty of Education Science, University of Granada, 18071 Granada, Spain; felixzo@ugr.es (F.Z.-O.); galeanorojasdilan24@gmail.com (D.G.-R.); 2Faculty of Natural Resources, Escuela Superior Politécnica de Chimborazo, Riobamba 060150, Ecuador; dtenelanda@espoch.edu.ec; 3School of Dentistry, Universidad Nacional de Chimborazo, Riobamba 060150, Ecuador; 4Departamento de Educación e Innovación, Universidad Católica de Temuco, Temuco 4780000, Chile; mcresp@uct.cl; 5Instituto del Deporte, Universidad de Las Américas, Santiago 9170022, Chile

**Keywords:** quality of life, lifestyles, confinement, sex, adults, COVID-19

## Abstract

The state of confinement during the COVID-19 pandemic affected the quality of life of the general population. This study aims to define and contrast an explanatory model of the quality of life in adults and to analyze the relationships between these variables based on the state of confinement and sex. A total of 872 people from Chile aged between 17 and 50 (M = 21.70 years; SD= 3.272), of both sexes (60.90% male and 39.1% female) participated in this research, of whom 46.6% were not confined when tested and 53.4% were confined, analyzing the quality of life. A model of multi-group structural equations was performed, which adjusted very well (χ^2^ = 559.577; DF = 38; *p* < 0.001; IFC = 0.934; NFI = 0.916; IFI = 0.934; RMSEA = 0.061). The results show a positive and direct relationship among all the variables studied and the structural equation model proposed according to confinement and sex reveals a good fit in all the evaluation indexes. Stress and sleep, personality, and introspection were the indicators with the greatest influence in the four models, followed by the family and friends indicator with a medium correlation strength, such as the health monitoring dimension, although this was not as influential in confined individuals. The main conclusions are that the best adjustments are obtained in confined adults and females, and the data show that the psychological indicators obtained (stress and sleep, personality, and introspection) have the greatest influence on adults in the four models proposed with regard to their quality of life.

## 1. Introduction

Studies on well-being and quality of life have increased in the last decade and are now a priority topic for the study of young people [1,2], adults [3,4,5] and the older population [6,7]. They are distributed across all countries of the world [8,9] and are the main focus of most policies in various world contexts. Above all, during the COVID-19 pandemic, the collapse of health systems and confinement measures that generated greater vulnerability and a propensity for health to deteriorate [1] demonstrated the need to strengthen well-being systems and policies. Various factors were contemplated at a psychological, physical, social, and environmental level that exert great influence on the quality of life of the population [7].

Identifying the problem is the first step toward reducing harmful habits and behavior, followed by creating behavioral patterns that should be addressed and worked on from an early age so they are acquired by adulthood [10,11]. In this regard, authors such as Hewitt et al. [12], Flores et al. [13] and Peterson et al. [4], focus their work on the university population, understanding that this is when emancipation from the family home occurs the most and where acquired harmful behaviors (excessive consumption of alcohol or drugs) increase.

Family and friends in the stages prior to adulthood are considered priority issues for human beings. In this sense, the university stage is characterized by young people beginning to live independently, separating from their nuclear family [14]. Due to the characteristics of this stage, where individuals are forced to meet their needs related to cleanliness and eating, among others, there are numerous studies that focus their analysis on this population [2,11,12,15]. The family exerts a special influence on young people, since after overcoming the adolescent stage, in which the subject moves away from the family nucleus, prioritizing social relations with their peer groups [16]. After this stage, family relationships improve due to the new role the young person acquires; therefore, the family influences the behavioral habits of young people [10,17,18].

One of the key psychological factors in human development related to developing quality of life is stress; some studies find a correlation between low levels of quality of life and increased stress, rising further during periods of COVID-19 [8,19,20,21] due to fear of infection and the related consequences (contagions or deaths).

Multiple studies have discussed the beneficial effects of proper nutrition on life expectancy and quality of life [22,23], as it has a positive effect on public health, reducing the risk of suffering from cardiovascular diseases, Alzheimer’s, depression, diabetes, and cancer, among others [24,25].

Introspection and personality are two elements that intervene directly in the state of well-being in such a way that adults with a better self-concept, self-esteem, or motivation manifest high levels of quality of life reflected in emotionally stable and long-lasting behaviors with greater participation and social interaction [26,27].

The benefits of physical activity in preventing and treating a variety of chronic diseases, such as obesity, hypertension, and diabetes, among others [28,29], are supported by numerous studies [24,30,31], giving physical activity a major role in the control of morbidity and premature mortality caused by chronic noncommunicable diseases [32,33]. Although, as Hootman et al. [34], O’Regan et al. [35] and Walsh et al. [36] pointed out, the decline in physical activity in adulthood should not be overlooked. Another notable aspect of this parameter is the significant contribution of physical activity to the development of the dimensions of human personality, including emotional intelligence, self-concept or social factors that help to relate to other people, among others [37,38].

Based on the scarcity of studies that address quality of life in times of pandemic, the following study is proposed with the objectives of defining and contrasting an explanatory model of quality of life and well-being in adults, and analyzing the existing associations between family and friends, physical activity, nutrition, sleep and stress, alcohol and drugs, personality, introspection, health control and other behaviors, depending on whether they were subjected to lockdown as a COVID preventive confinement and based on sex, through a multi-group structural equation analysis.

## 2. Materials and Methods

### 2.1. Design and Participants

A total of 872 adults aged between 17 (university students close to turning 18 years old) and 50 (students with work obligations who carry out university studies in the evening) (M = 21.70 years; SD = 3.272) participated in this descriptive and cross-sectional research study with non-probabilistic convenience sampling. We found that 531 (60.90%) were males and 341 (39.1%) females, where 406 (46.6%) were not confined and 466 (53.4%) were confined (Table 1). The selection was performed by sampling for convenience, considering the inclusion criteria of being enrolled in higher education of technical or professional career programs taught in the daytime modality, and in the case of people in total confinement, not suffering from any type of pathology that would prevent them from participating in the study; these were the criteria for inclusion and exclusion. The sample was obtained from different public and private higher education institutions in Chile, technical schools, or professional training institutions in the Metropolitan Region of Chile, requesting the participation of all the centers that volunteered to collaborate. It is necessary to indicate that 87 questionnaires were excluded after detecting that they were incorrectly filled out or that data were missing, that is, in principle, there were 959 participants.

### 2.2. Variables and Instruments

An ad hoc sociodemographic questionnaire with open and closed questions was used to collect information on sex (male and female), age, pathologies (yes or no) and, if any, the type of pathology. 

The Spanish version of the FANTASTIC questionnaire was used to analyze the lifestyles of adults. This questionnaire has been validated by Sharratt et al. [39], translated into Spanish by Ramírez-Vélez and Agredo [40] and adapted in Chile by Barrón et al. [41].

The questionnaire consists of 30 items that are scored using a 3-option Likert-type scale. The instrument presents a model composed of ten dimensions: family and friends, physical activity, nutrition, tobacco, alcohol and drugs, sleep and stress, personality, introspection, health control, and other behaviors (the tobacco dimension was eliminated because it did not provide adequate reliability values). In this study, a reliability of α= 0.87 was obtained.

### 2.3. Procedure

Adults were contacted (via institutional email) by the University of Granada (Spain) and the University of Santiago de Chile (Chile) to inform them of the nature of the study, and participants participated in the research on a voluntary basis. The informed consent document was then administered, and the questionnaires were completed using Google Forms. Furthermore, the anonymity of the participants was guaranteed by clarifying that the use of the data collected would be for scientific purposes only. This research is conducted under the ethical committee of the University of Granada code 641/CEIH/2018.

### 2.4. Data Analysis

For the analysis of the basic descriptions, the statistical software IBM SPSS^®^ version 22.0 for Windows was used. IBM AMOS^®^ 23 was used to analyze the relationships between the constructs involved in the structural model. After developing the theoretical model, a path analysis was carried out, considering the relationships of the matrix from a multi-group analysis, grouping the participants into confined or unconfined as a grouping variable. Finally, two different structural models were configured to verify if the relationships between the variables studied vary according to the sex (male or female).

The path models are based on seven observable variables and one latent variable to determine the indicators (Figure 1). In the proposed models, causal explanations of the latent variables are formulated from the observed relationships between the indicators, considering the reliability of the measurements. Measurement errors are also included in the observable variables so they can be directly controlled. Unidirectional arrows are lines of influence between latent and observable variables, being interpreted as multivariate regression coefficients. Bidirectional arrows show the relationship between latent variables, also representing regression coefficients.

Lifestyle (LS) acts as an exogenous variable and is inferred by nine indicators: FAM (family and friends), PA (physical activity), NUT (nutrition), ALC (alcohol and drugs), S (sleep and stress), PER (personality), INT (introspection), HM (health monitoring) and OC (other behaviors).

Model fit was checked to verify the compatibility of the model and the empirical information obtained. The reliability of the fit was performed based on goodness-of-fit criteria [42].

## 3. Results

The proposed structural equation model for the lifestyle reveals a good fit in all the evaluation indices. The Chi-square shows a significant value of *p* (*χ*^2^ = 114.596; DF = 27; *p* < 0.001). However, standard interpretation cannot be applied to this index, in addition to the problem posed by its sensitivity to sample size. Thus, other standardized adjustment indices that are less sensitive to sample size are used. The comparative fit index (CFI) achieved a value of 0.934, which is excellent. The normalized fit index (NFI) achieved a value of 0.916 and the incremental fit index (IFI) was 0.934, both of which are excellent. The root mean square error of approximation (RMSEA) obtained an acceptable value of 0.061.

Figure 2 and Table 2 show the estimated values of the structural model parameters for the unconfined individuals. These values should be adequate in magnitude and the effects should be significantly different from zero. Also, improper estimates should not be obtained as negative variances and the values obtained indicate an acceptable model fit with IFC = 0.902, NFI = 0.878, IFI = 0.912 and RMSEA = 0.079.

Statistically significant relationships are observed at the *p* < 0.001 level between all lifestyle categories and their dimensions, which are positive and direct. The relationship between lifestyle of unconfined adults is significant at the *p* < 0.001 level, with all variables being positive and direct, showing a high strength of correlation with sleep and stress (r = 0.663), personality (r = 0.609), and introspection (r = 0.690), with these indicators exerting the most influence; family and friends (r = 0.498) and health control (r = 0.435) establish a medium correlation; while the rest of indicators exert an influence with lower correlation strength (r = 0.310 in physical activity; r = 0.297 in nutrition; r = 0.302 in alcohol and drugs and r = 0.278 in other behaviors). 

Figure 3 and Table 3 show the estimated values of the structural model parameters for confined adults. These should be adequate in magnitude and the effects should be significantly different from zero. Also, improper estimates should not be obtained as negative variances. The model values are acceptable (IFC = 0.944; NFI = 0.907; IFI = 0.945 and RMSEA = 0.053).

Statistically significant relationships are observed at the *p* < 0.001 level between some of the lifestyle categories and their dimensions, which are positive and direct. The relationship between the lifestyle of adults in confinement is significant at the *p* < 0.001 level, showing a high strength of correlation with sleep and stress (r = 0.757), personality (r = 0.659) and introspection (r = 0.803), with these indicators exerting the greatest influence. On the other hand, family and friends (r = 0.430) and physical activity (r = 0.439) established a medium correlation; while the rest of indicators exerted influence with lower correlation strength (r = 0.234 in nutrition and r = 0.271 in health control).

Figure 4 and Table 4 show the estimated values of the structural model parameters for male participants. They should be adequate in magnitude and the effects should be significantly different from zero. Also, improper estimates should not be obtained as negative variances, and the values obtained show acceptable adjustment indicators (IFC = 0.912, NFI = 0.878, IFI = 0.913 and RMSEA = 0.065).

Statistically significant relationships are observed at the *p* < 0.001 level between most of the lifestyle categories and their dimensions, with these being positive and direct. The relationship between lifestyle of the male sex is significant at the *p* < 0.001 level, with all variables being positive and direct, showing a high strength of correlation with sleep and stress (r = 0.718), personality (r = 0.638) and introspection (r = 0.752), with these indicators having the greatest influence; family and friends (r = 0.466), physical activity (r = 0.387) and health control (r = 0.383) establish a mean correlation; while the rest of indicators have lower correlation strength (r = 0.223 in nutrition and r = 0.182 in alcohol and drugs).

Figure 5 and Table 5 show the estimated values of the structural model parameters for female participants. They should be adequate in magnitude and the effects should be significantly different from zero. Also, improper estimates should not be obtained as negative variances, and the values obtained show very good indicators of fit (IFC = 0.968, NFI = 0.924, IFI = 0.969 and RMSEA = 0.043).

Statistically significant relationships are observed at the *p* < 0.001 level between most of the lifestyle categories and their dimensions, being positive and direct. The relationship between lifestyle in the female sex is significant at the *p* < 0.001 level, with all variables being positive and direct, showing a high strength of correlation with sleep and stress (r = 0.761), personality (r = 0.686), introspection (r = 0.802) and physical activity (r = 0.519), with these indicators exerting the greatest influence. On the other hand, family and friends (r = 0.472) and health control (r = 0.396) establish a medium correlation; while alcohol and drugs exert influence with lower correlation strength (r = 0.223).

## 4. Discussion

The explanatory model on the quality of life of adults by sex and state of confinement presents a good fit and can serve as a guideline to detect, prevent, and determine models and lifestyles around the world. Studies along the same lines have been carried out in other places, such as Morocco [43], Italy [44], and the United Kingdom [45], with similar results, providing new perspectives that can be the basis for proposing recommendations, modifications and political and social interventions in relation to public services and the quality of life of the population.

The quality of life and the well-being of people are some of the actions that social-educational policies should follow to promote healthy lifestyles. In this regard, a systematic review and meta-analysis were conducted to support advice on weight loss by health staff [46]. In another papers, the relationships between lifestyle and relevant psychological aspects (mental health, depressive symptoms or stress) were studied, pointing out the importance of cognitive and physical exercise for improving the quality of life [47,48,49].

Low levels of sleep and stress and high levels of personality and introspection were associated with better quality of life [50]. The authors of this study emphasize the important role of cognitive and psychological factors in promoting the well-being of an individual. Srinivasan-Raj et al. [21] also pointed out these indicators as key elements in an overall strategy to improve the quality of life in adults.

Social support has a great influence on healthy habits and quality of life [51,52,53]. Family and friends have also been related to and studied with quality of life levels, showing positive associations [11,12,13,14,15,16,17,18,19,20,21,22,23,24,25,26,27,28,29,30,31,32,33,34,35,36,37,38,39,40,41,42,43,44,45,46,47,48,49,50,51,52,53,54]. In this study, the relationships have been positive and with average correlations emphasizing the importance of family and social relationships in a better state of well-being. As, stated by Ilic et al. [15], making friends and receiving praise and recognition from family and friends is key to adhering to a healthier lifestyle. 

Other recent studies confirm the importance of sports and health management in adults [55,56,57], including the importance of continued physical activity and regular health check-ups [35]. These aspects have also been the subject of research in confinement because of their possible relationship with physical inactivity [57,58], observing that the greater the sedentary behavior, the lower the quality of life and health indices [6,7].

Nutrition and alcohol and drug consumption also seem to influence the lifestyle and quality of life of adults, with better nutrition [58,59] and lower alcohol and drug consumption being associated with better overall well-being [6,52,60].

The proposed quality of life model obtained better adjustments in confined and female people. This is favored by the characteristics of the situation, as in the case of confinement, people assimilated the correct guidelines for the pathology and how to prevent it, and in the case of the female sex, there was a high percentage of housewives and mothers who stayed at home.

The data obtained reflect the fact that the most influential indicators of quality of life of the four models proposed are psychological factors (stress and sleep, personality and introspection). It is surprising that in confined people, there is an increase in these values with regard to the non-confined males and females. We suspect that confinement affects the cognitive level, as unknown situations are generated and cause moments of uncertainty, which some people face during situations of stress [61,62], or the situations bring out their personality. The results in this regard indicate that control of emotions should be encouraged as a protective element against psychologically harmful behaviors [63].

Family and friends obtained similar values in the four proposed groups, reaching the highest values in the group of non-confined people. We suggest that this is caused by the importance given by human beings to family and social relationships [11,52,53].

The study establishes a clear relationship between adults and health controls, being important in the non-confined, male, and female groups, but not in the group of people who were confined. This is understood to be facilitated by the cause of confinement being COVID-19, and that adults stopped worrying about other pathologies to some extent, focusing exclusively on this pathogen. 

Physical activity was considered highly important by the confined female and male sex, but not by the non-confined. The results indicate that policies to raise awareness of physical activity among women have had a positive impact and that women attach great importance to it to attain a better lifestyle and quality of life [64]. Conversely, in people with a daily life where labor and social factors prevail (non-confined people), the practice of physical activities is not very important for the improvement of the quality of life. It seems that the period of confinement has made people increase their appreciation of physical activity as an important element in improving the well-being of the individual [65,66].

The valuation of the remaining parameters does not vary considerably in the four groups, determining that the opinion held on nutrition, alcohol and drugs, or other behaviors is the same, in terms of their role within the quality of life of Chilean adults.

### Limitations

The study has some limitations. First, it is a cross-sectional study that does not establish cause-effect relationships, and second, we cannot generalize the data obtained to the whole confined adult population, as the time period of the study was only two months, compared to the non-confined population that has been in this situation for years. In this sense, as future lines of research, it would be interesting to conduct post-COVID comparisons, and identify the changes presented in the quality of life parameters analyzed in the present study. Furthermore, it will be interesting to analyze how the public health policies that were modified during the pandemic have been maintained, and how this affects the quality of life of individuals today.

## 5. Conclusions

The main conclusions of this study are that the model of the lifestyle in adults, based on sex and state of confinement, presents a good fit. Similarly, the best fit is obtained in confined adults and female adults.

It was concluded that stress and sleep, personality, and introspection were the indicators with the greatest influence in the four models for confined, unconfined, male, and female subjects. In addition, the family and friends indicator presented a medium correlation strength in all models, such as the health control dimension, although this was not as influential in confined individuals, and physical activity, which had a greater influence in confined individuals and males.

In this sense, the data obtained show that psychological indicators have the greatest influence on adults of the four proposed models in terms of their quality of life. Therefore, it is necessary to reconsider the way of addressing these indicators and generate preventive, support, and strengthening measures to improve this and all other dimensions of the lifestyle addressed in this study, with the aim of achieving a better quality of life in the general population.

## Figures and Tables

**Figure 1 healthcare-12-00774-f001:**
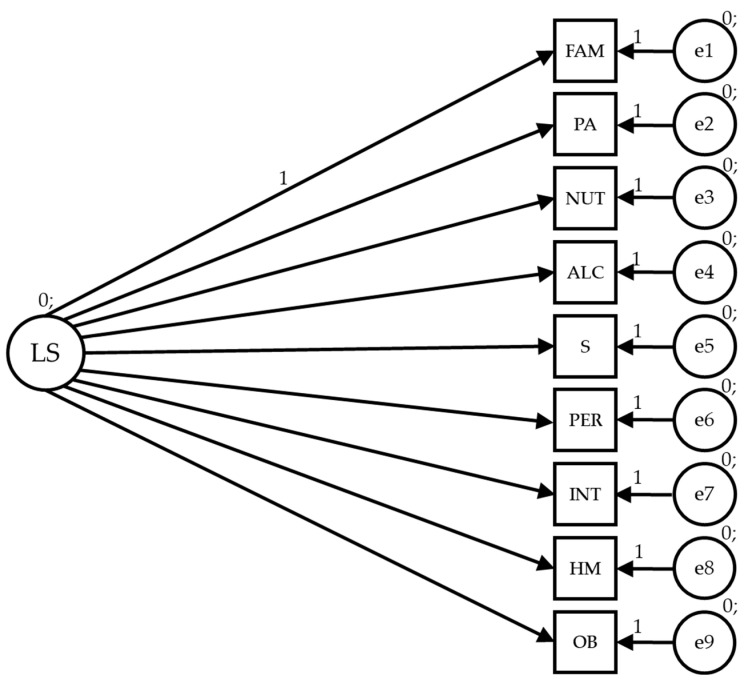
Theoretical model of lifestyle dimensions.

**Figure 2 healthcare-12-00774-f002:**
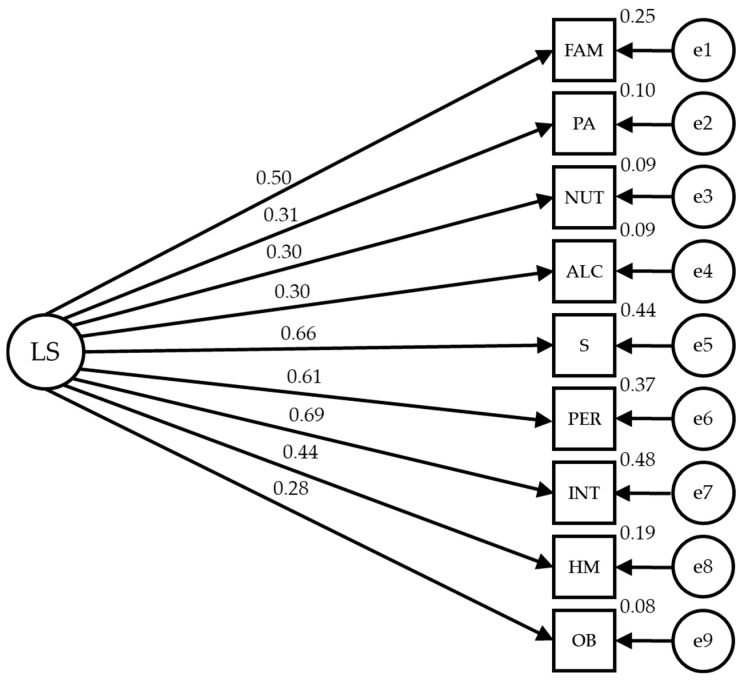
Model of structural equations for unconfined individuals.

**Figure 3 healthcare-12-00774-f003:**
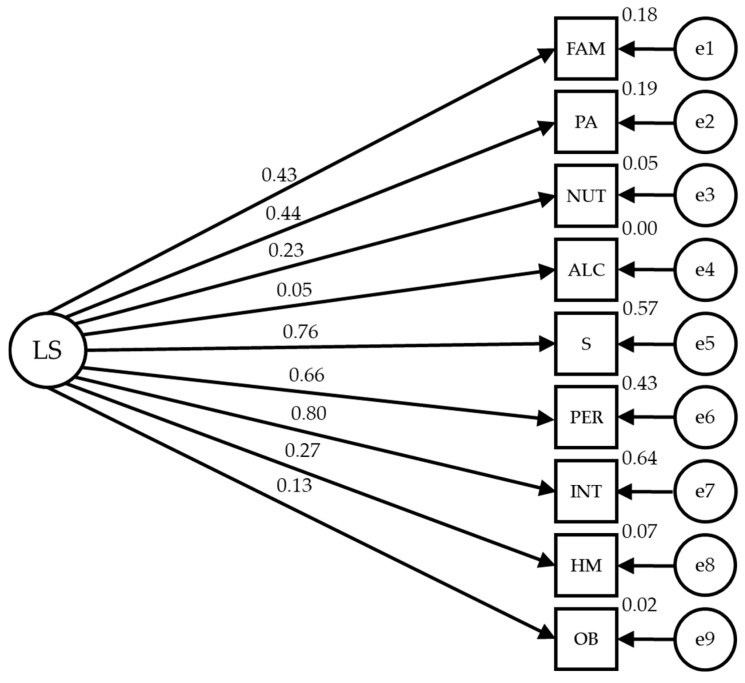
Model of structural equations for confined individuals.

**Figure 4 healthcare-12-00774-f004:**
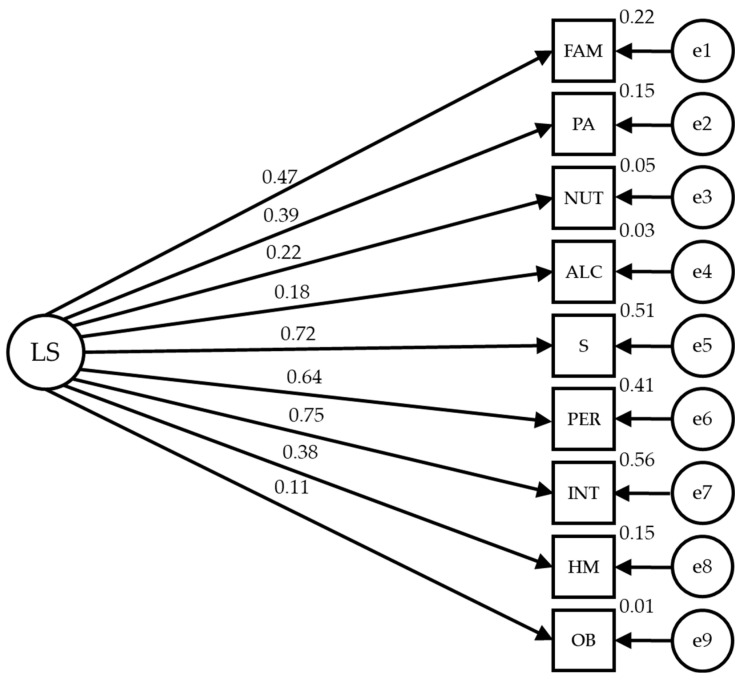
Male structural equation model.

**Figure 5 healthcare-12-00774-f005:**
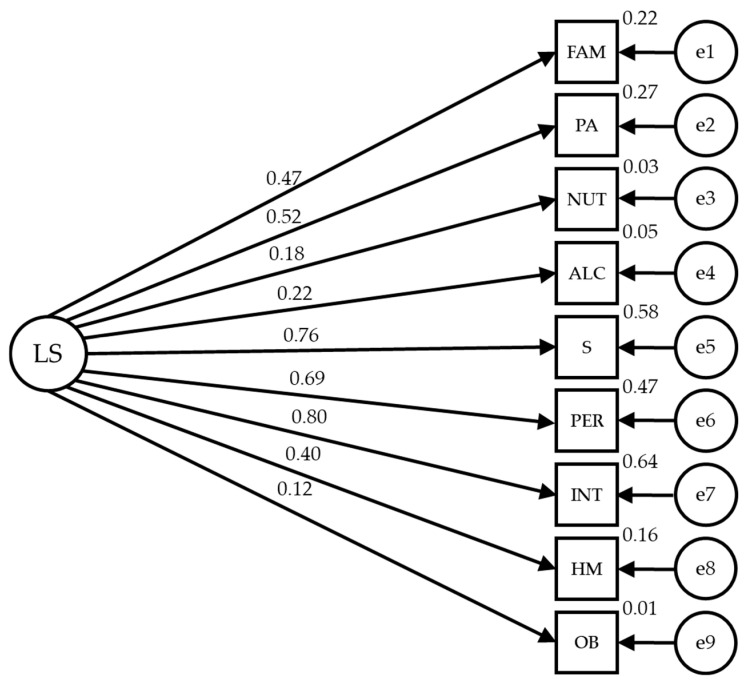
Female structural equation model.

**Table 1 healthcare-12-00774-t001:** Study descriptions.

			Confinement Status		Total
			Yes	No	
Sex	Male	Count	254	277	531
		% Sex	47.8%	52.2%	100.0%
		% Confinement	54.5%	68.2%	60.9%
	Female	Count	212	129	341
		% Sex	62.2%	37.8%	100.0%
		% Confinement	45.5%	31.8%	39.1%
Total		Count	466	406	872
		% Sex	53.4%	46.6%	100%
		% Confinement	100%	100%	100%

**Table 2 healthcare-12-00774-t002:** Structural model for unconfined adults.

Relationships between Variables			R.W.				S.R.W.
			Estimates	E.E.	C.R.	*p*	Estimates
FAM	←	LS	1.002	0.113	7.631	***	0.498
PA	←	LS	0.580	0.118	4.923	***	0.310
NUT	←	LS	0.551	0.116	4.752	***	0.297
ALC	←	LS	0.489	0.101	4.822	***	0.302
S	←	LS	1.203	0.149	8.064	***	0.663
PER	←	LS	1.037	0.134	7.762	***	0.609
INT	←	LS	1.119	0.137	8.186	***	0.690
HM	←	LS	0.902	0.142	6.356	***	0.435
OB	←	LS	0.484	0.108	4.493	***	0.278

Note1: LS, lifestyle; FAM, family and friends; PA, physical activity; NUT, nutrition; ALC, alcohol and drugs; S, sleep and stress; PER, personality; INT, introspection; HM, health monitoring and OB, other behaviors. Note2: R.W., regression weights; S.R.W., Standardized Regression Weights; E.E., Error Estimation; C.R., Critical Ratio. Note3: *** *p* < 0.001.

**Table 3 healthcare-12-00774-t003:** Structural model for confined adults.

Relationships between Variables			R.W.				S.R.W.
			Estimates	E.E.	C.R.	*p*	Estimates
FAM	←	LS	1.490	0.026	7.622	***	0.430
PA	←	LS	0.984	0.149	6.594	***	0.439
NUT	←	LS	0.375	0.090	4.184	***	0.234
ALC	←	LS	0.050	0.050	1.000	0.317	0.051
S	←	LS	1.682	0.201	8.388	***	0.757
PER	←	LS	1.247	0.155	8.028	***	0.659
INT	←	LS	1.670	0.197	8.482	***	0.803
HM	←	LS	0.464	0.099	4.707	***	0.271
OB	←	LS	0.199	0.082	2.420	0.016	0.128

Note1: LS, lifestyle; FAM, family and friends; PA, physical activity; NUT, nutrition; ALC, alcohol and drugs; S, sleep and stress; PER, personality; INT, introspection; HM, health monitoring and OB, other behaviors. Note2: R.W., regression weights; S.R.W., Standardized Regression Weights; E.E., Error Estimation; C.R., Critical Ratio. Note3: *** *p* < 0.001.

**Table 4 healthcare-12-00774-t004:** Structural model in adult males.

Relationships between Variables			R.W.				S.R.W.
			Estimates	E.E.	C.R.	*p*	Estimates
FAM	←	LS	0.984	0.076	7.622	***	0.466
PA	←	LS	0.803	0.121	6.636	***	0.387
NUT	←	LS	0.373	0.087	4.272	***	0.223
ALC	←	LS	0.245	0.069	3.557	***	0.182
S	←	LS	1.407	0.152	9.243	***	0.718
PER	←	LS	1.111	0.126	8.842	***	0.638
INT	←	LS	1.346	0.144	9.358	***	0.752
HM	←	LS	0.703	0.107	6.578	***	0.383
OB	←	LS	0.185	0.080	2.306	0.021	0.115

Note1: LS, lifestyle; FAM, family and friends; PA, physical activity; NUT, nutrition; ALC, alcohol and drugs; S, sleep and stress; PER, personality; INT, introspection; HM, health monitoring and OB, other behaviors. Note2: R.W., regression weights; S.R.W., Standardized Regression Weights; E.E., Error Estimation; C.R., Critical Ratio. Note3: *** *p* < 0.001

**Table 5 healthcare-12-00774-t005:** Structural model in female adults.

Relationships between Variables			R.W.				S.R.W.
			Estimates	E.E.	C.R.	*p*	Estimates
FAM	←	LS	0.954	0.078	7.622	***	0.472
PA	←	LS	1.191	0.178	6.676	***	0.519
NUT	←	LS	0.303	0.102	2.962	0.003	0.184
ALC	←	LS	0.254	0.072	3.535	***	0.223
S	←	LS	1.520	0.190	8.003	***	0.761
PER	←	LS	1.258	0.163	7.693	***	0.686
INT	←	LS	1.617	0.199	8.123	***	0.802
HM	←	LS	0.738	0.132	5.613	***	0.396
OB	←	LS	0.198	0.098	2.019	0.044	0.122

Note1: LS, lifestyle; FAM, family and friends; PA, physical activity; NUT, nutrition; ALC, alcohol and drugs; S, sleep and stress; PER, personality; INT, introspection; HM, health monitoring and OB, other behaviors. Note2: R.W., regression weights; S.R.W., Standardized Regression Weights; E.E., Error Estimation; C.R., Critical Ratio. Note3: *** *p* < 0.001.

## Data Availability

Data available on request due to privacy and ethical restrictions.

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
