# Peer review of "Association between Quality of Life, Confinement, and Sex in Adults: A Multigroup Structural Equation Analysis"

_healthcare, 2024, doi:10.3390/healthcare12070774_

Round 1

Reviewer 1 Report

Comments and Suggestions for Authors

The article "Association between Quality of Life, Confinement, and Sex in Adults: A Multigroup Structural Equation Analysis" explores the impact of confinement on adults' quality of life, with a focus on gender differences. It employs multigroup structural equation modeling to analyze data from 872 Chilean participants. The study aims to establish and compare a model explaining adults' quality of life and examines the relationships among these variables based on confinement status and gender, offering insights into the psychological effects of confinement in different demographic groups.

1. In the Design and Participants section, indicate “that 87 questionnaires were excluded after detecting that they were incorrectly filled out or that data were missing”. However, in Table 1 the sample description still shows a total of 872 participants. Review and clarify

2.  There are two typos in sections 2.3 and 2.4:

-    2.3. . Procedure

-    2.4. . Data analysis

3. In section 2.3, it is stated, "Adults were contacted by the University of Granada (Spain) and the University of Santiago de Chile (Chile) to inform them of the nature of the study, and participants were selected to participate in the research on a voluntary basis." It would be advisable to specify the method or means through which the participants were contacted. 4. In the statement of the objective, for the analysis of the associations, not all the indicators that are subsequently part of the analysis are mentioned. Review “and analyze the existing associations between family and friends, physical activity, nutrition, stress, personality, introspection and health.”

5. The Figures 2, 3, 4 and 5 present indicators in Spanish, they do not match with the presentation of Figure 1 or the language of the article (English). Review

6. Unify language, for example: health control (HC) in the tables and health monitoring (HM) in the text.

7. Check that the acronyms in Tables 2, 3, 4 and 5 correspond to the concept, for example: O.C: Other Behaviors, P.R.: Regression Weights, P.E.R.: Standardized Regression Weights. They seem to be the acronym for the Spanish translation.

8.    It would be advisable to propose directions for future research.

Author Response

Dear the editor and reviewers,

We would like to express our gratitude for the time taken to review this manuscript and for the comments made, which we believe to be critical for producing rigorous and quality research. We have detailed below the changes made in the original article: “Association between Quality of Life, Confinement, and Sex in Adults: A Multigroup Structural Equation Analysis” (healthcare-2863130).

Modifications have been made in the original manuscript following the reviewers’ comments. For each modification we have written: the original comment as written by the reviewer in addition to the page and line number; and the change made in response to that comment. Changes have been made using the tool “Track changes” enabling editor and reviewers to identify modifications easily.

MODIFICATIONS

EDITOR:

Comment 1:

The article "Association between Quality of Life, Confinement, and Sex in Adults: A Multigroup Structural Equation Analysis" explores the impact of confinement on adults' quality of life, with a focus on gender differences. It employs multigroup structural equation modeling to analyze data from 872 Chilean participants. The study aims to establish and compare a model explaining adults' quality of life and examines the relationships among these variables based on confinement status and gender, offering insights into the psychological effects of confinement in different demographic groups.

  1. In the Design and Participants section, indicate “that 87 questionnaires were excluded after detecting that they were incorrectly filled out or that data were missing”. However, in Table 1 the sample description still shows a total of 872 participants. Review and clarify

Response 1:

We accept the recommended modifications.

Regarding the sample and the 87 excluded questionnaires, the answer is given in line 95 y 96. The excluded questionnaires were part of an initial sample of 959 participants. Therefore, Table 1 describes the 872 participants who are part of the study and the subsequent analysis.

Comment 2:

  1. There are two typos in sections 2.3 and 2.4:

-    2.3. . Procedure

-    2.4. . Data análisis

Response 2:

Have been corrected.

Comment 3:

  1. In section 2.3, it is stated, "Adults were contacted by the University of Granada (Spain) and the University of Santiago de Chile (Chile) to inform them of the nature of the study, and participants were selected to participate in the research on a voluntary basis." It would be advisable to specify the method or means through which the participants were contacted.
  2. In the statement of the objective, for the analysis of the associations, not all the indicators that are subsequently part of the analysis are mentioned. Review “and analyze the existing associations between family and friends, physical activity, nutrition, stress, personality, introspection and health.”

Response 3:

The means of contact with the participants was through institutional email. (line 111)

In lines 74 and 75, mention is made of the missing indicators in the objective statement.

Comment 4:

The Figures 2, 3, 4 and 5 present indicators in Spanish, they do not match with the presentation of Figure 1 or the language of the article (English). Review

Response 4:

The figures have been modified with their respective acronyms in English.

Comment 5:

Unify language, for example: health control (HC) in the tables and health monitoring (HM) in the text.

Response 5:

Regarding the unification of the language between the text, tables and figures, the same terminology is used: HM: health monitoring.

Comment 6:

Check that the acronyms in Tables 2, 3, 4 and 5 correspond to the concept, for example: O.C: Other Behaviors, P.R.: Regression Weights, P.E.R.: Standardized Regression Weights. They seem to be the acronym for the Spanish translation.

Response 6:

In Tables 2, 3, 4 and 5, the acronyms corresponding to the English translation of the concepts were assigned: OB: Other Behaviors, RW.: Regression Weights, S.R.W.: Standardized Regression Weights.

Comment 7:

It would be advisable to propose directions for future research.

Response 7:

In the limitations section, mention is also made of future lines of research.

Reviewer 2 Report

Comments and Suggestions for Authors

Thank you for inviting me to review the manuscript entitled "Association between Quality of Life, Confinement, and Sex in Adults: A Multigroup Structural Equation Analysis" (manuscript ID: healthcare-2863130) submitted to Healthcare.

This study presents a comprehensive analysis of the relationships between Quality of Life, Confinement, and Sex in Adults. This research is capable of contributing significantly to the literature. Although it is well planned and written, there are certain aspects that need authors’ attention. My observations are presented below:

        (1)     The study title is good.

        (2)     The abstract requires minor modifications. I suggest improving interpretations of the results for readability and comprehension.

        (3)     The introduction also needs minor revision. I suggest the authors strengthening the study rationale by emphasizing the theoretical, practical, and policy implications. 

        (4)     The methods section also needs minor improvements. The authors should mention the participants’ recruitment method. Moreover, the inclusion and exclusion criteria should also be mentioned.

        (5)     The results section also need slight amendments. For example, the statistical values should be reported following a standard guideline such APA.

        (6)     The discussion section needs slight improvements. The interpretations of the study findings should be carried out in continuation of the research questions. Detailed theoretical, research and practical implications of the study results can be added. Novel findings should be highlighted. I strongly suggest adding subheadings for limitations, directions for future research and conclusion. Conclusion should not be the repetition of the results but it should include the reflections of the main findings.

        (7)     The references are ok.

In short, the study represents a good research needing minor amendments. Given the minor amendments, the study will surely contribute the field of knowledge. The manuscript may be considered for acceptance provided that it undergoes minor spell check, language editing and modifications given above.

I thank the editor again for providing me with the opportunity to review the manuscript.

With best regards,

Reviewer

Comments on the Quality of English Language

Minor language editing is suggested.

Author Response

Dear the editor and reviewers,

We would like to express our gratitude for the time taken to review this manuscript and for the comments made, which we believe to be critical for producing rigorous and quality research. We have detailed below the changes made in the original article: “Association between Quality of Life, Confinement, and Sex in Adults: A Multigroup Structural Equation Analysis” (healthcare-2863130).

Modifications have been made in the original manuscript following the reviewers’ comments. For each modification we have written: the original comment as written by the reviewer in addition to the page and line number; and the change made in response to that comment. Changes have been made using the tool “Track changes” enabling editor and reviewers to identify modifications easily.

MODIFICATIONS

EDITOR:

Comment 1:

Thank you for inviting me to review the manuscript entitled "Association between Quality of Life, Confinement, and Sex in Adults: A Multigroup Structural Equation Analysis" (manuscript ID: healthcare-2863130) submitted to Healthcare.

This study presents a comprehensive analysis of the relationships between Quality of Life, Confinement, and Sex in Adults. This research is capable of contributing significantly to the literature. Although it is well planned and written, there are certain aspects that need authors’ attention. My observations are presented below:

The study title is good.

Comment 2:

The abstract requires minor modifications. I suggest improving interpretations of the results for readability and comprehension.

Response 2:

Modifications were made to the abstract, describing the main results more clearly.

Comment 3:

The introduction also needs minor revision. I suggest the authors strengthening the study rationale by emphasizing the theoretical, practical, and policy implications. 

Response 3:

Modifications have been made to more clearly state the political and social implications of the study.

Comment 4:

The methods section also needs minor improvements. The authors should mention the participants’ recruitment method. Moreover, the inclusion and exclusion criteria should also be mentioned.

Response 4:

The recruitment method and the inclusion and exclusion criteria have been mentioned between lines 86 and 90. In addition, the procedure section mentions the means through which the study participants were contacted.

Comment 5:

The results section also need slight amendments. For example, the statistical values should be reported following a standard guideline such APA.

Response 5:

The presentation of the statistical values in the results section has been modified.

Comment 6:

The discussion section needs slight improvements. The interpretations of the study findings should be carried out in continuation of the research questions. Detailed theoretical, research and practical implications of the study results can be added. Novel findings should be highlighted. I strongly suggest adding subheadings for limitations, directions for future research and conclusion. Conclusion should not be the repetition of the results but it should include the reflections of the main findings.

Response 6:

The implications mentioned in the discussion have been delved into and the recommendations regarding the limitations and conclusions section have also been followed.

Comment 7:

The references are ok.

Reviewer 3 Report

Comments and Suggestions for Authors

This manuscript examines the impact of covid-lockdown policies on self reported quality of life for males and females. Using data on 872 respondents collected from Chile, this paper uses structural equation modeling to examine the impact of confinement on quality of life measures. 

This manuscript has a number of strengths. It adds to the international literature on the impact of covid restrictions on health. As the authors note, similar studies have examined data from countries such as the United Kingdom and Morocco. In addition, it adds to studies on quality of life during pandemics. The study uses a number of quality of life measures, including the role of physical activity, nutrition, stress, and the role of family and friends. In addition it comapares effectively between male and female respondents, The study addresses a need for additional research on quality of life during pandemics. The analytical strategy is approach to the questions explored in the data. It effectively employs SEM to test a multidimensional model of these relationships.  The model is well constructed and is consistent with existing literature.

However, for the manuscript to be satisfactory a number of issues need to be addressed by the authors.

First, it is not clear from the title or abstract or first page that the focus of the study is on the impact of covid restrictions  on quality of life. This needs to be clarified earlier in the manuscript, perhaps by changing the title or revising the abstract accordingly.

Second, the authors make reference (line 82) to the fact that the study is cross sectional. However, on line 121 the authors make reference to "grouping the participants into pre-COVID and post-COVID as a grouping variable." This is unclear. Does this mean that two separate sets of data were collected. Does this mean that data were collected from two separate time points (pre covid and post-COVID)? This is how it apoears. This must be clarified by the authors. Related to this, its not clear at what time point(s) the data actually were collected. During what month(s)/years were the data actually collected? The authors need to specify this. If the authors had explicitly stated this, it may help clear up this confusion. If there are indeed two separate time points, as it appears (pre and post covid), this needs to be clearly reflected in the analyses presented. There is a coment on line 301 that the data collection took two months. Assuming then that this is cross sectional, the comment referred to on line 121 about precovid and post-covid appears to be an error.

Substantial editing is required. Editing is needed for clarity, to address grammatical issues, and to address typographical errors.  These are listed by line number below:

Line 22: The sentence that begins "As main conclusions, it is determined that the..." Should read something like this: "The best adjustments are obtained in confined adults..." This would allow the authors to be clearer and avoid passive voice ("it is determined").  Alternatively, something like this would be possible: "The main conclusions are that the best adjustments...."

Line 34: The first sentence should read something like this: "Identifying the prolem is the first step toward reducing harmful habits and behavior. followed by creating..."

Lines 42-43; The sentence that begins "In this sense, the university stage is characterized by... should read "In this sense the university stage is characterized by....separating fron their nuclear family." The word "family" is missing.

Line 63: The first sentence should read "The benefits of physical activity in prevening...among others [28-29] are supportec by..."

Line 69: The sentence that begins "Another notable aspect..." should read "Another notable aspect of this parameter is the significant contribution of physical activity..."

Line 95: The word 'confinement" is repeatedly missspelled in Table 1. This needs to be corrected.

Lines 96-106: This section is confusing and contains a fragment "FANTASTIC questionnaire." This is unclear. What does this refer to? What questionnaire did the authors use? Was it one developed by Sharratt et al and translated into Spanish? If so, the authors need to say that. The first sentence, that begins "Sociodemographic questionnaire, where sex..." is not a complete sentence. This needs to be rewritten. This entire section needs to be carefully edited and rewritten. 

Lines 107-115: There is a stray period before the section title. This needs to be cleaned up. 

Line 112: The sentence that begins "The researchers were present during data collection...." is unclear. How did this occur? Were questionnaires given out to groups of students? Were participants visited by researchers and given individual questionnaires? The authors need to clarify what this means. 

Line 133: The title is not clear. It should be something more descriptive such as "Theoretical Model of Lifestyle indicators."

Line 143: The first sentence should read "The proposed structural equation model for the practice of..." Which model is this? Is this model 2? Also, is it a model of physical activity for unconfined individuals or confined individuals? For that matter,  if there is a model just for physical activity, are there models for each of the other indicators such as NUT (nutrition)? If so, where are they? Or, is this sentence incorrect and there is not a model for the effects of physical activity presented in this paper? This must be clarified.

Line 157: Figure 2 is unreadable as presented and needs to be redrawn or reprinted. 

Line 178: Same issue for Figure 3. As presented it is also unreadable. 

Line 198: Same issue for Figure 4.

Line 218: Same issue for Figure 5.

Line 243: The sentence that begins "In another review and papers, the relationships..." needs to be rewritten. Its awkwardly written. 

Line 248-49: The sentence that begins "The authors of this study emphasize....role of cognitive and psychological factors in..." The word "that" should be replaced by the word "of."

Lines 269-276: There are several issues to address here. The first sentence should read "The data obtained reflect the fact that the most influential indicators of quality of life of the four models proposed are psychological factors (stress and sleep), and it is surprising that...regard to the nonconfined males and females." 

Line 272: The sentence that begins "The authors consider that confinement.." should read something like this "We suspect that confinement affects the cognitive level, as..."

Line 279: The sentence that begins; "The authors consider.." should read something like this "We suggest that this is..." 

Comments on the Quality of English Language

The manuscript needs substantial editing for English grammar, spelling, and clarity. These issues are noted in the comments above.

Author Response

Dear the editor and reviewers,

We would like to express our gratitude for the time taken to review this manuscript and for the comments made, which we believe to be critical for producing rigorous and quality research. We have detailed below the changes made in the original article: “Association between Quality of Life, Confinement, and Sex in Adults: A Multigroup Structural Equation Analysis” (healthcare-2863130).

Modifications have been made in the original manuscript following the reviewers’ comments. For each modification we have written: the original comment as written by the reviewer in addition to the page and line number; and the change made in response to that comment. Changes have been made using the tool “Track changes” enabling editor and reviewers to identify modifications easily.

MODIFICATIONS

EDITOR:

Comment 1:

This manuscript examines the impact of covid-lockdown policies on self reported quality of life for males and females. Using data on 872 respondents collected from Chile, this paper uses structural equation modeling to examine the impact of confinement on quality of life measures.

This manuscript has a number of strengths. It adds to the international literature on the impact of covid restrictions on health. As the authors note, similar studies have examined data from countries such as the United Kingdom and Morocco. In addition, it adds to studies on quality of life during pandemics. The study uses a number of quality of life measures, including the role of physical activity, nutrition, stress, and the role of family and friends. In addition it comapares effectively between male and female respondents, The study addresses a need for additional research on quality of life during pandemics. The analytical strategy is approach to the questions explored in the data. It effectively employs SEM to test a multidimensional model of these relationships.  The model is well constructed and is consistent with existing literature.

However, for the manuscript to be satisfactory a number of issues need to be addressed by the authors.

First, it is not clear from the title or abstract or first page that the focus of the study is on the impact of covid restrictions  on quality of life. This needs to be clarified earlier in the manuscript, perhaps by changing the title or revising the abstract accordingly.

Response 1:

Modifications were made to the abstract and the introduction section to provide greater clarity regarding the study topic.

Comment 2:

Second, the authors make reference (line 82) to the fact that the study is cross sectional. However, on line 121 the authors make reference to "grouping the participants into pre-COVID and post-COVID as a grouping variable." This is unclear. Does this mean that two separate sets of data were collected. Does this mean that data were collected from two separate time points (pre covid and post-COVID)? This is how it apoears. This must be clarified by the authors. Related to this, its not clear at what time point(s) the data actually were collected. During what month(s)/years were the data actually collected? The authors need to specify this. If the authors had explicitly stated this, it may help clear up this confusion. If there are indeed two separate time points, as it appears (pre and post covid), this needs to be clearly reflected in the analyses presented. There is a coment on line 301 that the data collection took two months. Assuming then that this is cross sectional, the comment referred to on line 121 about precovid and post-covid appears to be an error.

Substantial editing is required. Editing is needed for clarity, to address grammatical issues, and to address typographical errors.  These are listed by line number below:

Line 22: The sentence that begins "As main conclusions, it is determined that the..." Should read something like this: "The best adjustments are obtained in confined adults..." This would allow the authors to be clearer and avoid passive voice ("it is determined").  Alternatively, something like this would be possible: "The main conclusions are that the best adjustments...."

Line 34: The first sentence should read something like this: "Identifying the prolem is the first step toward reducing harmful habits and behavior. followed by creating..."

Lines 42-43; The sentence that begins "In this sense, the university stage is characterized by... should read "In this sense the university stage is characterized by....separating fron their nuclear family." The word "family" is missing.

Line 63: The first sentence should read "The benefits of physical activity in prevening...among others [28-29] are supportec by..."

Line 69: The sentence that begins "Another notable aspect..." should read "Another notable aspect of this parameter is the significant contribution of physical activity..."

Line 95: The word 'confinement" is repeatedly missspelled in Table 1. This needs to be corrected.

Lines 96-106: This section is confusing and contains a fragment "FANTASTIC questionnaire." This is unclear. What does this refer to? What questionnaire did the authors use? Was it one developed by Sharratt et al and translated into Spanish? If so, the authors need to say that. The first sentence, that begins "Sociodemographic questionnaire, where sex..." is not a complete sentence. This needs to be rewritten. This entire section needs to be carefully edited and rewritten.

Lines 107-115: There is a stray period before the section title. This needs to be cleaned up.

Line 112: The sentence that begins "The researchers were present during data collection...." is unclear. How did this occur? Were questionnaires given out to groups of students? Were participants visited by researchers and given individual questionnaires? The authors need to clarify what this means.

Line 133: The title is not clear. It should be something more descriptive such as "Theoretical Model of Lifestyle indicators."

Line 143: The first sentence should read "The proposed structural equation model for the practice of..." Which model is this? Is this model 2? Also, is it a model of physical activity for unconfined individuals or confined individuals? For that matter,  if there is a model just for physical activity, are there models for each of the other indicators such as NUT (nutrition)? If so, where are they? Or, is this sentence incorrect and there is not a model for the effects of physical activity presented in this paper? This must be clarified.

Line 157: Figure 2 is unreadable as presented and needs to be redrawn or reprinted.

Line 178: Same issue for Figure 3. As presented it is also unreadable.

Line 198: Same issue for Figure 4.

Line 218: Same issue for Figure 5.

Line 243: The sentence that begins "In another review and papers, the relationships..." needs to be rewritten. Its awkwardly written.

Line 248-49: The sentence that begins "The authors of this study emphasize....role of cognitive and psychological factors in..." The word "that" should be replaced by the word "of."

Lines 269-276: There are several issues to address here. The first sentence should read "The data obtained reflect the fact that the most influential indicators of quality of life of the four models proposed are psychological factors (stress and sleep), and it is surprising that...regard to the nonconfined males and females."

Line 272: The sentence that begins "The authors consider that confinement.." should read something like this "We suspect that confinement affects the cognitive level, as..."

Line 279: The sentence that begins; "The authors consider.." should read something like this "We suggest that this is..."

Response 2:

Indeed, the comment referred to in line 121 has been an error, the study corresponds, as mentioned in the design and participants section, to a cross-sectional design, with data collection carried out during a single period of two months.

All grammatical issues and typographical errors have been corrected and clarified in each section. We appreciate the comments offered, they have been of great help to us.
